# Decoding Visual Responses: Insights into Chronic Migraine and Medication Overuse Headache with Electrophysiological Analysis

**DOI:** 10.3390/jcm13206070

**Published:** 2024-10-11

**Authors:** Gianluca Coppola, Francesco Casillo, Gabriele Sebastianelli, Chiara Abagnale, Cherubino Di Lorenzo, Antonio Di Renzo, Mariano Serrao, Vincenzo Parisi

**Affiliations:** 1Department of Medico-Surgical Sciences and Biotechnologies, Sapienza University of Rome Polo Pontino ICOT, 04100 Latina, Italy; gianluca.coppola@uniroma1.it (G.C.); francesco.casillo@uniroma1.it (F.C.); gabriele.sebastianelli@uniroma1.it (G.S.); chiara.abagnale@uniroma1.it (C.A.); cherub@inwind.it (C.D.L.); mariano.serrao@uniroma1.it (M.S.); 2IRCCS–Fondazione Bietti, 00198 Rome, Italy; antoniomp777@hotmail.it

**Keywords:** medication overuse, chronic migraine, visual cortex, amplitude, medication withdrawal

## Abstract

**Background/Objectives**: Habituation and sensitization are opposite phenomena that play a role in the pathophysiology of episodic migraine and its progression to chronic migraine (CM). There have been just a few studies that have investigated these phenomena in patients with medication overuse headache (MOH) in comparison to those with chronic migraine (CM) and healthy controls (HCs), and the findings have been inconsistent. **Methods**: We measured and examined visual evoked potentials (VEPs) in 81 patients with MOH and 24 patients with CM, as well as 24 HCs. The VEPs were used to assess sensitization by analysing the amplitude of the first block (100 sweeps) and to evaluate habituation by measuring the amplitude response decrement after six blocks. We further examined patients diagnosed with MOH based on their acute medication type and after a 3-week acute medication withdrawal program. **Results**: There were no significant differences between groups in terms of the first N1-P1 VEP amplitude block and its habituation. It was found that patients with MOH had a greater drop in the amplitude of the VEP P1-N2 complex after repeated stimulation than patients with CM or HC. The VEP parameters showed no significant differences based on the specific overused drug and after a 3-week acute medication withdrawal. **Conclusions**: We propose that the results obtained in patients with MOH indicate an abnormal activation of inhibitory circuits in the parieto-occipital region in response to repeated modulatory stimuli.

## 1. Introduction

Within a year, up to 3% of patients with episodic migraine evolve into patients with chronic migraine (CM) [1]. One of the causative factors of transformation is the overuse of symptomatic drugs, which can lead to the diagnosis of medication overuse headache (MOH) [2]. All drugs commonly used for acute treatment can induce MOH. Among the hypothesized mechanisms underlying the transformation are central sensitization and a defect in pain inhibition by the brainstem and cortical systems [3]. Addictive behaviour and elevated relapse rates post-withdrawal may be attributed to the hypofunction of the orbitofrontal cortex [4]. Central sensitization arises from adaptive modifications in circuits governing ascending sensory transmission, leading to an overall enhancement of both nociceptive and innocuous afferent signals [5,6]. Certain studies suggest that central sensitization in chronic migraine and medication overuse headache predominantly involves areas beyond the trigeminovascular system, which is involved in the perception of pain. For instance, CM patients had the lowest phosphene threshold and suppression index, assessed using the magnetic suppression of perceptual accuracy tests and TMS-based methodologies to determine visual cortex excitability [7]. In a functional MRI study, persons with MOH demonstrated alterations in functional connectivity across multiple cortical networks, mostly affecting the visual regions, compared to healthy controls [8]. Visual cortical regions are significantly engaged in the processing of salient information, such as the experience of pain [9]. In a study presenting nociceptive laser stimuli alongside visual stimuli, the authors demonstrated that the sensorimotor processing of visual stimuli can be impaired by an automatic reallocation of attention to the nociceptive input [10]. The perception of tonic pain can alter visual evoked cortical responses in healthy individuals, but not in those with migraine [11]. Little is known about the direct effect of acute medications at the cortical level, including visual areas [3].

Electrophysiological studies showed a specific direct effect of individual drug classes at the level of the sensorimotor system, which in turn depends on the duration of the chronic and/or overuse phase [12]. In particular, the level of basal cortical excitability (sensitization) in both CM and MOH patients, as measured by the amplitude of evoked potentials, is increased in comparison to healthy subjects and then decreases late in CM (a phenomenon called transient sensitization, with a normal delayed habituation), whereas it continues to increase in MOH (called persistent sensitization, with a delayed lack of habituation). This is accompanied by changes in thalamocortical activity and the degree of lateral intracortical inhibition. In fact, while in patients with CM the level of thalamocortical activity and the degree of lateral intracortical inhibition is within normal limits, both are increased in patients with MOH compared to healthy subjects [12,13].

Contradictory results were obtained when analysing cortical responses to visual stimuli. Evidence in favour of central sensitization came from the analysis of visual responses recorded using a magneto-electroencephalography, which, similar to sensorimotor responses, were only initially transiently sensitized in CM patients [14]. After preventive treatment with topiramate, the sensitization of the visual cortex is replaced by an electrophysiological pattern similar to that of episodic interictal migraine, characterized by a tendency to decrease initial evoked responses followed by a delayed habituation deficit [15]. Other researchers, recording visual evoked potentials with an electroencephalograph in a small cohort of patients with CM and with CM+MOH patients, observed no between groups differences in the levels of sensitization and amplitude habituation [16,17]. Therefore, further research in a larger cohort of patients is needed to verify whether visual responses are different in patients with MOH and CM compared to healthy subjects [3].

The aim of this study is to record visual pattern evoked potentials (VEPs) in a group of patients with MOH and to compare them with a group of patients with CM and healthy controls (HCs). The primary outcome of this study is to test whether there are differences between the groups with regard to both the degree of initial sensitization and that of delayed habituation; the secondary outcome is to test whether these electrophysiological variables change depending on the type of medication the patient is overusing, and after 3 weeks off acute medication without the simultaneous start of preventive therapy. Based on previous results with somatosensory potentials, we hypothesize that MOH patients show persistent sensitization patterns, which are maximal in patients with excessive drug combination use.

## 2. Materials and Methods

The diagnosis of the patients was based on the criteria of the latest version of the International Classification of Headache Disorders [2].

### 2.1. Study Participants

Among consecutive patients attending the authors’ headache clinic, 100 provided informed consent to participate in the study, of whom 19 were excluded because of the inclusion and exclusion criteria. Participants were included if they were between 18 and 65 years of age and had at least a 1-year clinical history of migraine. Participants were excluded from the study if they were regularly taking medication (e.g., antibiotics, corticosteroids, antidepressants, benzodiazepines, or prophylactic migraine medication) during the 3 months preceding the study, except for contraceptive pills (taken by 5 HCs, 8 MOH, and 4 CM). All the participants underwent an ophthalmological assessment that involved determining the best-corrected visual acuity, examining the eye with a slit-lamp biomicroscope, measuring intraocular pressure, and conducting indirect ophthalmoscopy. 

Ophthalmological exclusion criteria: The presence of central scotoma, square-wave jerks, saccadic intrusions, and nystagmus in the primary point of gaze that could affect the capacity to maintain a stable fixation throughout the VEP recordings; coexistence of various systemic diseases (such as diabetes, systemic hypertension, and rheumatologic disorders) that could impact retinal function; and the presence of glaucoma or other illnesses affecting the cornea, lens (LOCS III stage < 1), uvea, or retina.

General exclusion criteria: Individuals with a history of other neurological disorders, systemic hypertension, diabetes, or other metabolic or autoimmune disease, or any other type of primary or secondary headache, were also excluded. 

Patients did not always experience the headaches on the same side. All participants received a complete description of the study and provided written informed consent. The study was approved by the local ethics review board and was conducted in accordance with the Helsinki Declaration.

According to the inclusion/exclusion criteria, the final dataset comprised 81 patients (Table 1), of whom 16 were diagnosed with de novo MOH (IHCD-III code 8.2) and 24 with de novo CM, with no history of medication overuse (ICHD-III code 1.3). MOH patients never underwent a detoxification program. The sample of patients with MOH included 21 patients overusing triptans (IHCD-III code 8.2.1), 32 overusing non-opioid analgesic drugs (NOAs) (IHCD-III code 8.2.3), and 28 patients overusing combinations of multiple drug classes, not individually overused (IHCD-III code 8.2.6). Before progressing to MOH, all patients had a clear-cut history of episodic migraine without aura (ICHD-III code 1.1). Because of the high number of headache days experienced by these patients, we decided to accept recordings non-exclusively during the pain-free phase, but even during a mild headache (1–5 on VAS scale). Because MOH patients tend to take acute medications compulsively and frequently during the day, it was impossible to prevent them from taking medication on the day of recordings. It was managed, however, to perform the recordings at least 3 h after the last medication intake. 

Of the 81 patients with MOH studied, 33 (10 triptan, 15 NOAs, and 8 combination overusers) agreed to be re-evaluated clinically and electrophysiologically (VEP recordings) 3 weeks after withdrawing from acute medication overuse, without prophylactic medication. The post-withdrawal recording session took place at least 72 h before and after an eventual migraine attack, as checked by telephone interview.

For comparison, VEPs were recorded in 24 healthy controls (HCs) with comparable age and sex distribution (Table 1), and no personal or familial history (first degree relatives) of migraine and no overt medical condition. To avoid variability due to hormonal changes, female participants were examined outside their pre-menstrual or menstrual cycles.

### 2.2. Recording of Visual Evoked Potentials (VEPs)

We conducted VEP recordings using the methodology described in our previously published research [18].

Subjects were seated in a semi-dark, acoustically isolated room in front of the display and surrounded by a uniform field of luminance of 5 cd·m^2^ for monocular recordings. We used a visual stimulus of a full-screen checkerboard pattern (contrast 80%, mean luminance 110 cd/m^2^) generated on a monitor and reversed in contrast at the rate of 3.1 reversals per second. At the viewing distance of 114 cm, in the monitor screen subtending 23 degrees, the checked edges subtended 15′ of the visual angle for the VEP recordings [19].

VEPs were derived from right monocular stimulation. VEPs were recorded from the scalp using silver/chloride cup electrodes placed at Oz (active electrode) and Fz (reference electrode, 10/20 system) [20]. A grounding electrode was placed on the right forearm.

Signals were amplified by Digitimer^TM^ D360 (bandwidth 0.05–2000 Hz, gain 1000) and recorded using a CED^TM^ power 1401 device (Cambridge Electronic Design Ltd., CED, Cambridge, UK). Six hundred consecutive traces, each lasting 200 msec, were collected and sampled at 4000 Hz. The cortical responses were divided into 6 sequential blocks of 100, consisting of at least 95 artifact-free traces. The off-line averaging of the responses in each block (“block averages”) was performed using Signal^TM^ software version 4.11 (CED Ltd., Cambridge, UK). Artifacts were automatically removed using Signal^TM^’s artifact rejection tool only if the signal amplitude exceeded 90 percent of the analogue-to-digital conversion (ADC) range, which was further checked through visual inspection. The EP signal was corrected off-line for DC deviations, eye movements, and blinking.

VEP components were identified according to their implicit times: N1 was defined as the major negative peak between 60 and 90 msec, P1 as the major positive peak following N1 between 80 and 120 msec, and N2 as the major negative peak following P1. 

We measured the N1, P1, and P2 implicit times and the peak-to-peak amplitude of the N1-P1 and of P1-N2 complexes (in mV).

Sensitization was defined as an increased N1-P1 amplitude recorded during block 1 (after a low number of 100 stimuli). Habituation was defined as the slope of the linear regression line for the 6 VEP blocks. Positive values indicate a lack of amplitude habituation (delayed augmenting responses), whereas negative values indicate a more significant amplitude habituation (delayed decreasing responses).

All recordings were collected by the researchers (G.S., C.A.), who had not met the participants before the examination and were not involved in recruiting and including the subjects. All recordings were numbered anonymously and analysed off-line blinded by a researcher, who was not blinded to the order of the blocks (G.C.).

### 2.3. Statistical Analysis

Sample size calculation was not based on formal statistics but on the previous literature [16,17]. We used the Statistical Package for the Social Sciences (SPSS) for Windows, version 21.0, for all analyses. Levene’s test was used to check for normal distribution, and all the considered variables displayed a normal distribution. For patients’ clinical features, variables were tested in a one-way analysis of variance (ANOVA) with factor groups of “subjects” (MOH, CM, and HCs). To assess behavioural changes in VEP amplitude between blocks 1 and 6, N1-P1 and P1-N2 amplitudes were tested first with a repeated-measure ANOVA for the factor group “subjects” and repeated measures “block” factors, then using the group factor “MOH subgroups” (MOH-triptans, MOH-NOAs, MOH-combination, and HCs). A separate repeated-measure ANOVA was carried out to compare the electrophysiological variables before and after 3-week acute medication withdrawal (time x group x type of overused medication factors). Tukey’s test was used for post hoc analyses. A paired sample *t*-test was used to compare clinical variables (days with headache and number of tablets taken during the month preceding and succeeding the withdrawal program) before and after acute medication withdrawal. Pearson’s correlation coefficient was calculated to test correlations between block 1 VEP amplitude or VEP habituation slope and clinical data (disease duration, days with headache, number of tablets taken per month, duration of chronic headache). *p* values of less than 0.01 were considered reflecting statistical significance to compensate for the number of clinical variables.

## 3. Results

Analysable VEP recordings were obtained from all patients and HCs participating in the study. 

VEP implicit times of N1, P1, and N2 components were not different between groups (for each measure F(2,127), *p* > 0.05) (Table 2).

The ANOVA testing of N1-P1 VEP amplitude block averages disclosed a significant effect for the block factor (F(5,635) = 9.64, *p* = 0.017), but failed to disclose an effect for the group factor (F(2,127) = 0.14, *p* = 0.86) and for the interaction block*group (F(10,635) = 1.51, *p* = 0.13). Post hoc analysis did not reveal significant between groups changes in amplitude of block 1 N1-P1 VEP (F(2,127)= 0.97, *p* = 0.38) (Figure 1). The habituation slope did not differ between groups (F(2,127)= 0.97, *p*= 0.38).

ANOVA testing of P1-N2 VEP amplitude block averages failed to disclose a significant effect for the factor group (F(2,127) = 0.54, *p* = 0.58), but disclosed a significant effect for the factor block (F(5,635) = 7.06, *p* < 0.001) and for the block*group interaction (F(10,635) = 2.06, *p* = 0.026). Post hoc analysis revealed that the significance of the block*group interaction was due to a significant drop in amplitude of block 4 and block 6 as compared to block 1 in the MOH group (*p* < 0.0001 and *p* = 0.04, respectively). Post hoc analysis did not disclose any significance between group changes in the amplitudes of block 1 P1-N2 VEP (F = (2,127) = 0.25, *p* = 0.78) (Figure 1). The habituation slope did not differ between groups (F(2, 127) = 0.25, *p* = 0.78).

In patients with MOH, the monthly number of acute medication intake correlated positively with monthly days with headache (r = 0.397, *p* < 0.001) and the duration of chronic phase (r = 0.611, *p* < 0.001). None of the VEP electrophysiological variables correlated to the patients’ clinical features (Table 3).

### 3.1. VEPs in Patients Stratified According to the Overused Acute Medication

When we stratified the data for patients with MOH according to the class of overused drugs, i.e., triptans, NOAs or combinations, the ANOVA for VEP N1-P1 amplitudes in the various blocks showed the main effect caused by the block factor (F(5,505) = 7.65, *p* < 0.001), but not by the drugs factor (F(3,101) = 0.10, *p* = 0.96) or their interactions (F(15,505) = 0.91, *p* = 0.56). ANOVA for VEP P1-N2 amplitudes in the various blocks also showed the main effect caused by the block factor (F(5,505) = 7.81, *p* < 0.001) but not drugs factor (F(3,101) = 0.86, *p* = 0.47) or their interactions (F(15,505) = 1.01, *p* = 0.44). Post hoc analysis did not reveal significance between group changes in block 1 N1-P1 and P1-N2 VEP (F(3,101)= 0.80, *p* = 0.49; F(3,101) = 0.32, *p* = 0.81, respectively) (Figure 2).

### 3.2. VEPs before and after Acute Medication Withdrawal

One month after starting a 3-week acute medication withdrawal, we observed a significant reduction in the number of headache days (T0 = 24.6 ± 6.3; T1 = 1.8 ± 2.1; T0 vs. T1: t(1,26)= 15.6, *p* < 0.001) and number of acute medication intake (T0 = 30.8 ± 20.6; T1 = 1.4 ± 2.3; T0 vs. T1: t(1,26) = 6.8, *p* < 0.001).

When we compared the electrophysiological data before and after the 3-week acute medication withdrawal, the ANOVA testing of N1-P1 VEP amplitude block averages did not reveal a significant effect for the time factor (F(1,60) = 0.07, *p* = 0.789), the type of medication factor (F(2,60) = 0.18, *p* = 0.84), and for the interaction of time*type (F(2,60) = 0.04, *p* = 0.96). The ANOVA testing of P1-N2 VEP amplitude block averages did not disclose a significant effect for the time factor (F(1,60) = 0.13, *p* = 0.716), for the type factor (F(2,60) = 1.02, *p* = 0.37), and for the interaction of time*type (F(2,60) = 0.10, *p* = 0.90) (Figure 3).

In summary, the current results indicate that only patients with MOH exhibit anomalies in delayed visual information processing, as confirmed by the examination of the P1-N2 component of VEPs. No significant variations in the amplitude and habituation of VEPs were seen between the subgroups of overusers and healthy controls and following abrupt acute drug withdrawal.

## 4. Discussion

In the present study, we searched for VEP abnormalities in patients with CM and in patients with MOH before and after abrupt acute medication discontinuation and according to the overused drugs. We found that in patients with CM and in patients with MOH, the VEP N1-P1 amplitude of block 1, reflecting sensitization, and the VEP N1-P1 amplitude behaviour during stimulus repetition, reflecting habituation, did not differ from those of HCs. In patients with MOH, the VEP P1-N2 amplitude reduction during stimulus repetition was significantly more pronounced than that of both HCs and patients with CM. N1-P1 and P1-N2 block 1 VEP amplitude and VEP delayed amplitude habituation did not differ according to the overused drug (triptan, NOA, or combination). In patients with MOH after 3 weeks off acute medication, no difference was found in the N1-P1 and P1-N2 block 1 VEP amplitude and VEP delayed amplitude habituation. None of the VEP electrophysiological variables correlated to the patients’ clinical features.

This study attempted to record VEPs, an electrophysiological measure of the mass activity of visual cortical neurons, to verify the level of the initial cortical sensitization and of delayed habituation in a group of patients with CM and patients with MOH, without ongoing prophylactic medication.

Previous attempts to study visual cortical excitability in CM and MOH used both electro- and magneto-encephalography. Using standard pattern-reversal VEP, Viganò et al. did not detect significant differences in patients with CM or CM+MOH as regards first N1-P1 and P1-N2 amplitude block and habituation when compared with HCs [16,17]. In contrast to these results, others, recording visual evoked magnetic field (VEF) with a magneto-electroencephalograph, found an increase in the amplitude of the first P100m block in CM patients in comparison to healthy subjects and episodic migraineurs outside attacks [14]. The degree of habituation of VEF amplitude in CM patients was not different from that in healthy subjects. The apparent discrepancy in the results obtained by recording VEPs and VEFs could be due to methodological differences intrinsic to the different techniques used. In fact, VEPs reflect the mass activity of the entire visual pathway, including cortical and subcortical stations, whereas the P100m amplitude of VEFs exclusively reflects cortical activity tangential to the scalp, with the absence of the conducted volume effect and the so-called ‘paradoxical lateralisation’ effect present in VEPs [21]. These characteristics of the VEF signal may explain why it was better at capturing alterations in cortical sensitisation levels than the VEP technique.

Our current results obtained in CM patients were in line with previous studies using the VEP method: the N1-P1 and P1-N2 amplitudes for both the early block and the late habituation were superimposable to those of healthy subjects [16,17]. Multichannel scalp recordings have documented a generator in the striate and extrastriate areas for the N1 component, dorsal extrastriate cortex of the middle occipital gyrus for the early phase of the P1 component, and a more complex genesis of the N2 component, probably deep on the centro-parietal level [22]. Consequently, our findings suggest that the visual path transmitting information to the striate and extrastriate visual areas is normally engaged in patients with CM.

In the group of patients with MOH, but not in the HC and CM patient groups, in contrast to previous VEP studies [17], we detected a significant decrease in P1-N2 amplitude in the later response blocks (fourth and sixth) compared to the initial block. Despite this, we did not detect a significant difference between the groups in the degree of habituation as measured by the slope of the regression line calculated from the amplitude of the six blocks. 

Interestingly, in a previous study using trains of transcranial magnetic stimuli (TMS) delivered to the sensorimotor cortex in groups of CM and MOH patients, we found a paradoxical inhibitory cortical activity in response to an excitatory TMS paradigm in MOH patients than in CM patients and HCs [23]. Furthermore, in a recent study, Viganò and colleagues found a statistically significant modulatory effect of an inhibitory TMS paradigm only on delayed N1-P1 and P1-N2 VEP habituation levels and not on the initial response [17]. Overall, these previous findings in MOH patients, in addition to our observation of normal N1-P1 amplitude and its habituation and the late drop in P1-N2 amplitude, could be due to a combination of normal primary visual area activation with an abnormal activation of parieto-occipital inhibitory circuits in response to repeated modulatory stimuli in patients with MOH compared to those with CM. Furthermore, as the learning phenomena of habituation and sensitisation are elementary forms of activity-dependent synaptic plasticity [24], we believe our results advance existing knowledge in that they show an imbalance of synaptic mechanisms of short-term potentiation and depression slightly in favour of the latter in subjects with MOH. From the analysis of our data, it is not possible to speculate whether this result reflects a primary dysfunction of cortical arousal mechanisms in these patients or is a secondary protective mechanism to counteract the overstimulation of sensory systems by symptomatic drugs. Our research indicates that the mechanisms of short-term plasticity triggered by the recurrent presentation of visual stimuli are partially impaired in MOH patients compared to pure CM patients and healthy volunteers. The subtle divergent plasticity mechanisms observed in the two patient groups may suggest that MOH and CM, although presenting a comparable phenotype, engage in distinct neurophysiological learning processes, potentially linked to divergent pathophysiological mechanisms underlying migraine chronification. Moreover, although we did not identify an association between the consumption of acute drugs and the electrophysiological parameters, persistent exposure to acute medication usage may lead to modifications in short-term plasticity mechanisms.

We further expanded the study of VEPs in patients with MOH by also analysing the responses of subgroups of patients categorised according to overuse drug and after the 3-week drug withdrawal program. We did not find a drug-specific effect on VEPs (N1-P1 and P1-N2 amplitude and habituation) and furthermore did not find statistically significant differences in VEPs after 3 weeks of symptomatic drug de-addiction, without the establishment of prophylaxis. These results are apparently in contrast to our earlier observation obtained by recording somatosensory evoked potentials (SSEPs). In these cases, we detected an initial sensitisation of SSEPs only in patients who overused NSAIDs and combination drugs, but not in those who overused triptans, despite the fact that all subgroups showed an equal delayed habituation deficit [12]. This discrepancy may be explained either by a possible specific pharmacological effect on the parietal cortex of the drugs in question or by a greater involvement of the parietal cortex in the pain processing than the visual cortex.

As in all previously mentioned VEP studies in patients with CM and MOH [14,16,17], we found that electrophysiological parameters did not correlate with any clinical characteristics. This was despite the fact that we found that in the MOH group the number of drugs consumed in a month increased progressively as the number of days with headache/month and the duration of the chronic phase increased.

A major strength of this study is that it recruited a large cohort of patients with CM and MOH, with no current prophylactic therapy and no previous attempt of drug withdrawal. A limitation of the present study is that it did not collect information on the personality traits and psychiatric comorbidity of our patients. Previous studies have in fact found that these variables can influence VEP responses [25,26]. A further drawback of our investigation is the impossibility of restricting acute medication consumption on the day of data recording for MOH patients. This is due to the recognized compulsive behaviour and anticipatory anxiety exhibited by these patients [3,27].

## 5. Conclusions

This study revealed delayed visual information processing abnormalities verified by the analysis of the P1-N2 component of VEPs only in MOH patients. Further investigations by independent research groups are necessary to ascertain whether this electrophysiological finding can serve as a biomarker for the disease, specifically in differentiating MOH from CM. The former will also assist in personalizing the available pharmacological and non-pharmacological interventions. In the MOH group of patients, no significant differences in the amplitude and habituation of VEPs were found in subgroups of overusers before and after abrupt acute medication discontinuation, as compared to healthy subjects. Additional electrophysiological investigations are required to ascertain whether particular personality traits, the existence of systemic and central diseases in comorbidity, or unique genetic variants could possibly influence sensory processing in CM and MOH patients.

## Figures and Tables

**Figure 1 jcm-13-06070-f001:**
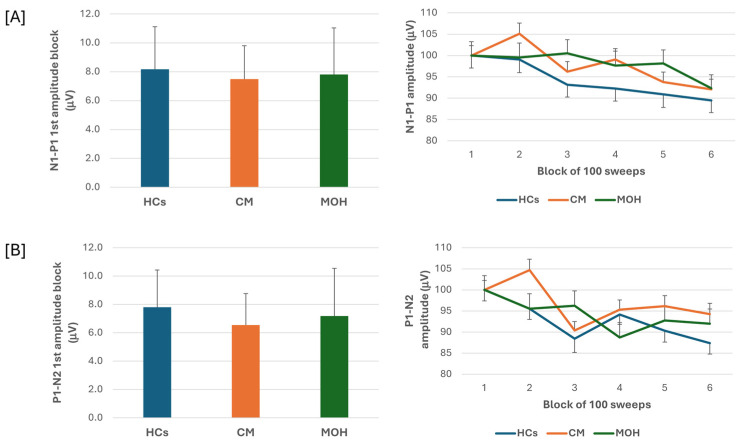
N1-P1 (**A**) and P1-N2 (**B**) visual evoked potential (VEP) mean amplitude of block 1 (100 averaged responses, **left** panel) and delayed amplitude habituation along six blocks (**right** panel) in healthy controls (HCs), patients with chronic migraine (CM), and patients with medication overuse headache (MOH).

**Figure 2 jcm-13-06070-f002:**
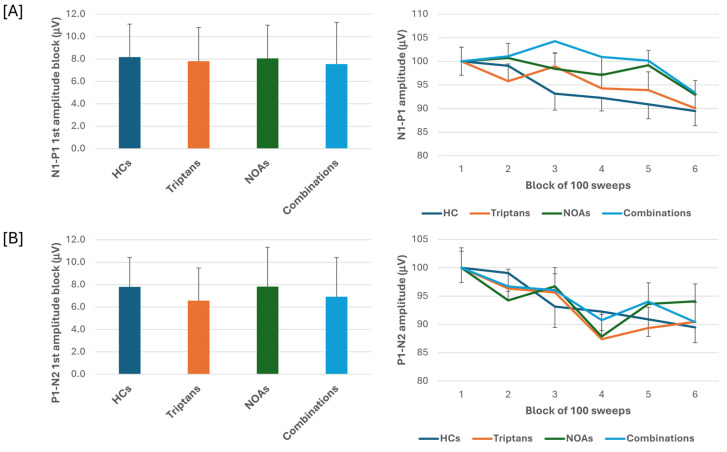
N1-P1 (**A**) and P1-N2 (**B**) visual evoked potential (VEP) mean amplitude of block 1 (100 averaged responses, **left** panel) and delayed amplitude habituation along six blocks (**right** panel) in subgroups of patients with medication overuse headache (MOH) overusing triptans and non-opioid analgesic drugs (NOAs) and of patients overusing combinations of multiple drug classes, although not individually overused.

**Figure 3 jcm-13-06070-f003:**
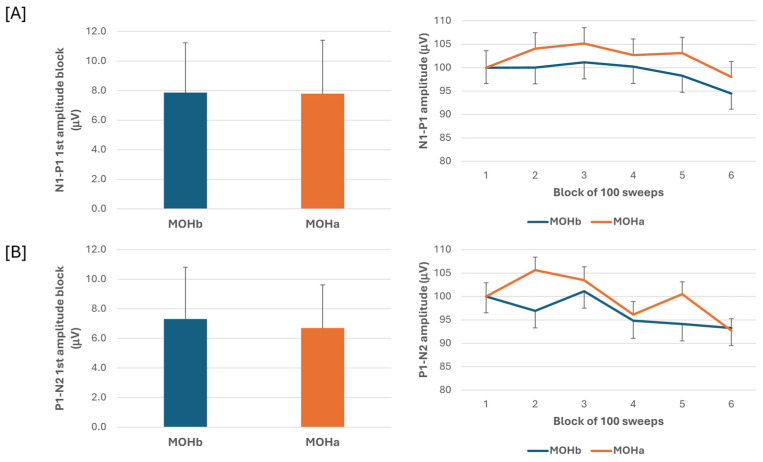
N1-P1 (**A**) and P1-N2 (**B**) visual evoked potential (VEP) mean amplitude of block 1 (100 averaged responses, **left** panel) and delayed amplitude habituation along six blocks (**right** panel) in a subgroup of patients with medication overuse headache (MOH) who underwent data recording before (MOHb) and after (MOHa) 3 weeks off medication.

**Table 1 jcm-13-06070-t001:** Demographic data of study participants and headache profiles of patients with chronic migraine (CM), with medication overuse headache (MOH-tot), and subgroups of triptan, NOA, and combination medication overusers. Data are expressed as mean ± SD. HC, healthy controls.

	HC (n = 24)	CM (n = 25)	MOH-tot (n = 81)	Triptan (n = 21)	NOA (n = 32)	Combination (n = 28)
Women (n)	18	19	68	17	30	21
Age (years)	44 ± 12	36 ± 13	42 ± 12	46 ± 8	38 ± 13	44 ± 12
Duration of history of migraine (years)		20.2 ± 14.0	24.5 ± 12.4	25.8 ± 11.4	22.3 ± 13.6	26.1 ± 11.8
Days with headache/month (n)		25.7 ± 6.1	23.9 ± 6.2	24.1 ± 5.9	24.1 ± 6.5	23.6 ± 6.2
Duration of the chronic phase (months)		23.2 ± 26.0	8.0 ± 22.3	10.5 ± 14.8	3.0 ± 3.2	11.6 ± 34.8
Tablet intake/month (n)		2.7 ± 3.2	35.8 ± 33.6	34.9 ± 32.4	31.0 ± 25.2	42.3 ± 42.5

**Table 2 jcm-13-06070-t002:** Visual evoked potential (VEP) N1, P1, and N2 implicit times in patients with chronic migraine (CM), with medication overuse headache (MOH-tot), and subgroups of triptan, non-opioid analgesic drug (NOA), and combination medication overusers. Data are expressed as mean ± SD. HC, healthy controls.

	HC (n = 24)	CM (n = 25)	MOH-tot (n = 81)	Triptan (n = 21)	NOA (n = 32)	Combination (n = 28)
VEP N1 (ms)	79.0 ± 7.9	74.3 ± 7.1	77.2 ± 6.8	78.9 ± 6.2	75.2 ± 6.6	78.4 ± 6.9
VEP P1 (ms)	107.4 ± 8.8	102.8 ± 7.8	105.4 ± 8.4	108.2 ± 9.5	102.7 ± 7.3	106.5 ± 8.1
VEP N2 (ms)	142.9 ± 12.3	142.6 ± 12.5	146.7 ± 12.6	144.9 ± 15.6	145.8 ± 11.1	149.2 ± 11.8

**Table 3 jcm-13-06070-t003:** Correlation analyses between the N1-P1 and P1-N2 first amplitude block and the habituation slopes with the clinical features of patients with chronic migraine (CM) and patients with medication overuse headache (MOH). After correction for the number of clinical variables, we did not find statistically significant correlations.

	CM (n = 25)				MOH (n = 81)			
	**First Block N1-P1**	**N1-P1 Habituation Slope**	**First Block P1-N2**	**P1-N2 Habituation Slope**	**First Block N1-P1**	**N1-P1 Habituation Slope**	**First Block P1-N2**	**P1-N2 Habituation Slope**
Duration of history of migraine (years)	r = 0.18 *p* = 0.40	r = 0.15 *p* = 0.47	r = 0.02 *p* = 0.91	r = - 0.06 *p* = 0.78	r = −0.04 *p* = 0.75	r = −0.03 *p* = 0.79	r = −0.05 *p* = 0.65	r = 0.04 *p* = 0.76
Days with headache/month (n)	r = −0.36 *p* = 0.08	r = −0.04 *p* = 0.85	r = −0.28 *p* = 0.16	r = 0.24 *p* = 0.23	r = −0.15 *p* = 0.18	r = 0.23 *p* = 0.04	r = −0.11 *p* = 0.35	r = 0.03 *p* = 0.77
Tablet intake/month (n)	r = −0.17 *p* = 0.44	r = −0.02 *p* = 0.93	r = −0.47 *p* = 0.02	r = 0.06 *p* = 0.79	r = −0.26 *p* = 0.02	r = 0.30 *p* = 0.04	r = −0.24 *p* = 0.03	r = 0.17 *p* = 0.14
Duration of the chronic phase (months)	r = −0.31 *p* = 0.30	r = 0.03 *p* = 0.92	r = −0.40 *p* = 0.17	r = −0.02 *p* = 0.94	r = −0.04 *p* = 0.71	r = 0.24 *p* = 0.04	r = −0.08 *p* = 0.52	r = −0.02 *p* = 0.87

## Data Availability

The data that support the findings of this study are available from the corresponding author upon reasonable request.

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
