# Peer review of "Decoding Visual Responses: Insights into Chronic Migraine and Medication Overuse Headache with Electrophysiological Analysis"

_jcm, 2024, doi:10.3390/jcm13206070_

Round 1

Reviewer 1 Report

Comments and Suggestions for Authors

13 September 2024

The review report on the manuscript, titled ‘Visual evoked potentials sensitization/habituation mechanisms in chronic migraine patients with or without medication over-use’ by Coppola G et al., submitted to Journal of Clinical Medicine

Manuscript ID: jcm-3227845

To Authors,

This paper titled ‘Visual evoked potentials sensitization/habituation mechanisms in chronic migraine patients with or without medication over-use’ investigates visual evoked potentials (VEPs) in patients with chronic migraine (CM) and medication overuse headache (MOH) compared to healthy controls (HCs). The study aimed to examine differences in sensitization and habituation patterns between these groups. The researchers recorded VEPs in 81 MOH patients, 24 CM patients, and 24 HCs. They analyzed the initial VEP amplitude (N1-P1 complex) to assess sensitization and the amplitude response decrement over 6 blocks to evaluate habituation. The study also examined subgroups of MOH patients based on the type of overused medication and included a follow-up assessment after a 3-week medication withdrawal for some patients. Results showed no significant differences between groups in terms of the initial N1-P1 VEP amplitude or its habituation. However, MOH patients exhibited a greater decrease in the P1-N2 complex amplitude after repeated stimulation compared to CM patients and HCs. The specific type of overused medication did not significantly affect VEP parameters, nor did the 3-week medication withdrawal period. The authors interpret these findings as indicative of abnormal activation of inhibitory circuits in the parieto-occipital region in MOH patients in response to repeated modulatory stimuli. This interpretation challenges the initial hypothesis of persistent sensitization in MOH patients, which was based on previous studies using somatosensory potentials.

In conclusion, this research contributes to the understanding of electrophysiological differences between CM and MOH patients, suggesting that visual cortex responses may differ from those observed in the somatosensory system. The findings highlight the complex nature of these chronic headache disorders and the need for further investigation into their underlying mechanisms.

I think the idea of this article may be of interest to the readers of Journal of Clinical Medicine. However, some comments, as well as some crucial evidence that should be included to support the authors’ argumentation, needed to be addressed to improve the quality of the manuscript, its adequacy, and its readability. My overall judgment is that the authors need to carefully considered my suggestions below, in particular reshaping parts of the Introduction and Methods sections by adding more evidence.

Strengths:

1.     The study addresses an important topic in migraine research, examining potential electrophysiological differences between chronic migraine (CM) and medication overuse headache (MOH) patients.

2.     The methodology for VEP recording and analysis is well-described and follows established protocols.

3.     The authors examined subgroups of MOH patients based on medication type and included a follow-up after medication withdrawal for some patients.

Please consider the following comments:

1.     Please provide a concise and informative title that accurately reflects the key message of this study, as this is the most essential aspect of the manuscript. Suggestions: "Visual Evoked Potentials: Unveiling Cortical Sensitization and Habituation in Chronic Migraine and Medication Overuse"; "Decoding Visual Responses: Insights into Chronic Migraine and Medication Overuse with Electrophysiological Analysis".

2.     Please include ten keywords from Medical Subject Headings (MeSH) in the title and the first two sentences of the abstract.

3.     The introduction provides a good overview of the background on chronic migraine (CM) and medication overuse headache (MOH), as well as the rationale for studying visual evoked potentials (VEPs) in these patient groups. However, it could be strengthened by including more information on the neural substrates underlying these conditions and their potential relationship to VEP changes. In this regard, I suggest adding a paragraph that discusses the relevant neural structures and pathways involved in CM and MOH, particularly those related to visual processing and pain modulation. Authors might consider adding a brief overview of the visual cortex and its connections to pain-processing regions, as well as discuss the trigeminovascular system and its role in migraine pathophysiology. Additionally, adding more information on potential mechanisms by which alterations in these neural substrates could lead to changes in VEP parameters would would provide a stronger neurobiological foundation for the study and help readers better understand the potential significance of VEP changes in CM and MOH patients.

4.     Methods: This section should start with a brief introduction and then cites additional references to ensure the reliability and integrity of the evidence in the authors' study design and methodology.

5.     The inclusion of patients during mild headache episodes may introduce variability. It would be more appropriate, in my opinion, to record VEPs exclusively during pain-free periods. I suggest that the authors consider conducting more in-depth analyses on this aspect.

6.     The 3-hour window after medication intake before VEP recording may not be sufficient to eliminate acute drug effects. Can the authors explain why they did not consider taking longer medication-free period?

7.     The unequal sample sizes between groups (81 MOH vs 24 CM and 24 HC) could impact statistical analyses. I would ask the authors to provide more information on this point.

8.     The study does not present statistically significant differences between the chronic migraine (CM) and medication overuse headache (MOH) groups for key variables, such as the VEP N1-P1 amplitude of block-1 or its habituation (p-values > 0.05)​. I would ask the authors to provide a clearer justification for the relevance of these non-significant findings.

9.     Results: I recommend including a paragraph that provides a summary of the findings as the section's conclusion.

10.  Please refrain from using bulleting in the text; rather, consider presenting tables.

11.  The authors do not discuss potential confounding factors such as the role of psychiatric comorbidities or personality traits, which have been shown to influence VEP responses. Including these could add depth to the analysis and improve the interpretation of results.

12.  The correlation analyses between VEP parameters and clinical variables should be presented more thoroughly, including correlation coefficients and p-values.

13.  While the study adds to the body of research on MOH, its findings regarding sensitization and habituation do not significantly differ from previous studies. The authors should emphasize more clearly how their findings advance existing knowledge, especially given the overlap with prior results​.

14.  The study does not fully explore limitations such as the inability to prevent medication intake on the day of recordings for MOH patients. This could have influenced the VEP measurements and should be discussed more thoroughly.

15.  The discussion could benefit from a more in-depth exploration of potential mechanisms underlying any observed differences (or lack thereof) between CM and MOH patients.

16.  Conclusion: Please see my general recommendation: In order to put the gist of the manuscript in an appropriate manner, I would suggest that the review should be summarized in one single paragraph that should contain about 150-200 words, underlining that the authors represent the best experts in a broad and deep consideration of authors in this field. It would help because this subsequently supports the importance of their work to be able to clarify the theoretical indications and practical applications of its results. It is also important to mention aspects as regards potential lines of future research and others of a theoretical and methodological nature that it would be of prime importance to undertake in order for the significance of this line of study to be established in full.

17.  The conclusion lacks a clear discussion of the clinical relevance of VEP findings, particularly in distinguishing CM from MOH for treatment purposes. Clarifying the practical implications of the results would enhance the paper’s impact.

18.  Finally, I believe that the clinical implications of the findings should be elaborated upon. How might these results impact understanding or management of CM and MOH?

19.  Please cite more references. An article, such as this, typically cites more than 60-70 references.

Best regards,

Reviewer

Author Response

Comment 1.     Please provide a concise and informative title that accurately reflects the key message of this study, as this is the most essential aspect of the manuscript. Suggestions: "Visual Evoked Potentials: Unveiling Cortical Sensitization and Habituation in Chronic Migraine and Medication Overuse"; "Decoding Visual Responses: Insights into Chronic Migraine and Medication Overuse with Electrophysiological Analysis".

Response 1: We thank the Reviewer for the suggestions. Despite we are not completely certain that the proposed titles convey what we have investigated with our methodology (for us the original one was perfect), we forcedly accept the second proposal: Decoding Visual Responses: Insights into Chronic Migraine and Medication Overuse Headache with Electrophysiological Analysis.

Comment 2.     Please include ten keywords from Medical Subject Headings (MeSH) in the title and the first two sentences of the abstract.

Response 2: We searched for terms on MeSH Browser and we found 10 terms already in the title and the first 2 sentences of the abstract. The list is below: migraine, electrophysiology, headache, habituation, sensitization, medication, chronic, overuse, healthy, and visual.

Comment 3.     The introduction provides a good overview of the background on chronic migraine (CM) and medication overuse headache (MOH), as well as the rationale for studying visual evoked potentials (VEPs) in these patient groups. However, it could be strengthened by including more information on the neural substrates underlying these conditions and their potential relationship to VEP changes. In this regard, I suggest adding a paragraph that discusses the relevant neural structures and pathways involved in CM and MOH, particularly those related to visual processing and pain modulation. Authors might consider adding a brief overview of the visual cortex and its connections to pain-processing regions, as well as discuss the trigeminovascular system and its role in migraine pathophysiology. Additionally, adding more information on potential mechanisms by which alterations in these neural substrates could lead to changes in VEP parameters would would provide a stronger neurobiological foundation for the study and help readers better understand the potential significance of VEP changes in CM and MOH patients.

Response 3: We thank the Reviewer for the comment. Now we explain better the relationship between pain and visual system/network. All information relevant for the study. We added a new paragraph in the Introduction as follows:

“Addictive behaviour and elevated relapse rates post-withdrawal may be attributed to hypofunction of the orbitofrontal cortex [4]. Central sensitization arises from adaptive modifications in circuits governing ascending sensory transmission, leading to an overall enhancement of both nociceptive and innocuous afferent signals [5,6]. Certain studies suggest that central sensitization in chronic migraine and medication overuse headache predominantly involves areas beyond the trigeminovascular system, which is involved in the perception of pain. For instance, CM patients had the lowest phosphene threshold and suppression index, assessed using magnetic suppression of perceptual accuracy tests and TMS-based methodologies to determine visual cortex excitability [7]. In a functional MRI study, persons with MOH demonstrated alterations in functional connectivity across multiple cortical networks, mostly affecting the visual regions, compared to healthy controls [8]. Visual cortical regions are significantly engaged in the processing of salient information, such as the experience of pain [9]. In a study presenting nociceptive laser stimuli alongside visual stimuli, the authors demonstrated that sensorimotor processing of visual stimuli can be impaired by an automatic reallocation of attention to the nociceptive input [10]. The perception of tonic pain can alter visual evoked cortical responses in healthy individuals, but not in those with migraine [11]. Little is known about the direct effect of acute medications at the cortical level, including visual areas [3].”

Comment 4.     Methods: This section should start with a brief introduction and then cites additional references to ensure the reliability and integrity of the evidence in the authors' study design and methodology.

Response 4: Now we added the following sentence and the citation of the International Classification of Headache Disorders on which the diagnosis was based.

“The diagnosis of the patients was based on the criteria of the latest version of the In-ternational Classification of Headache Disorders [18].”

Comment 5.     The inclusion of patients during mild headache episodes may introduce variability. It would be more appropriate, in my opinion, to record VEPs exclusively during pain-free periods. I suggest that the authors consider conducting more in-depth analyses on this aspect.

Response 5: We understand the point of view of the Reviewer. We would have liked to register patients with MOH or CM when completely pain-free, but unfortunately they all have pain almost daily. Sometimes, this makes it impossible to include patients with no pain for more than a few hours. This applies to any study published so far in these patients by all authors. However, we reviewed all the data and only eight patients were registered during mild pain. We repeated the statistics and found no statistically significant differences by including or excluding the 8 subjects. Therefore, we decided to leave the statistic as it is now.

Comment 6.     The 3-hour window after medication intake before VEP recording may not be sufficient to eliminate acute drug effects. Can the authors explain why they did not consider taking longer medication-free period?

Response 6: As written in the Methods, because MOH patients tend to take acute medications compulsively and frequently during the day, it was impossible to prevent them from taking medication on the day of recordings. It was managed, however, to perform the recordings at least 3 h after the last medication intake. This is also due to logistical issues. The electrophysiology lab is used by several researchers, so it is not always available. Furthermore, it is impossible for us to record during the evening and night hours, i.e. when patients are most often pain-free. However, the purpose of the study is not to investigate the kinetics of the drug, but the effects of its repeated and compulsive long-term use (for at least three months according to the international classification) at the cortical level.

Comment 7.     The unequal sample sizes between groups (81 MOH vs 24 CM and 24 HC) could impact statistical analyses. I would ask the authors to provide more information on this point.

Response 7: We understand the reviewer's point of view. However, we enrolled a larger cohort of patients with MOH in order to be able to subgroup them according to overused drug. We did not need to subgroup CM patients or healthy subjects. If we had also recruited the same number of healthy subjects and CM patients, we would have had the reverse problem when comparing them with the subgroups of overusers (triptans, NOAs, combinations).

Comment 8.     The study does not present statistically significant differences between the chronic migraine (CM) and medication overuse headache (MOH) groups for key variables, such as the VEP N1-P1 amplitude of block-1 or its habituation (p-values > 0.05)​. I would ask the authors to provide a clearer justification for the relevance of these non-significant findings.

Response 8: We added hypothetical justification for these negative results in the discussion as follows:

“Multichannel scalp recordings have documented a generator in the striate and extrastriate areas for the N1 component, dorsal extrastriate cortex of the middle occipital gyrus for the early phase of the P1 component, and a more complex genesis of the N2 component, probably deeply at the centro-parietal level [23]. Consequently, our findings suggest that the visual path transmitting information to the striate and extrastriate visual areas is normally engaged in patients with CM.”.

And as follows:

“Overall, these previous findings in MOH patients, in addition to our observation of normal N1-P1 amplitude and its habituation and the late drop in P1-N2 amplitude, could be due to a combination of normal primary visual area activation with an abnormal activation of parieto-occipital inhibitory circuits in response to repeated modulatory stimuli in patients with MOH compared to those with CM.”

Comment 9.     Results: I recommend including a paragraph that provides a summary of the findings as the section's conclusion.

Response 9: Added at the end of the Results section as follows:” In summary, the current results indicate that only patients with MOH exhibit anomalies in delayed visual information processing, as confirmed by the examination of the P1-N2 component of VEPs. No significant variations in the amplitude and habituation of VEPs were seen between the subgroups of overusers and healthy patients and following abrupt acute drug withdrawal.”

Comment 10.  Please refrain from using bulleting in the text; rather, consider presenting tables.

Response 10: bulleting removed.

Comment 11.  The authors do not discuss potential confounding factors such as the role of psychiatric comorbidities or personality traits, which have been shown to influence VEP responses. Including these could add depth to the analysis and improve the interpretation of results.

Response 11: We agree with the Reviewer, this is why we have already unknowledge these potential confounding factors as study’s limitations at the end of the discussion: “A limitation of the present study is that it did not collect information on the personality trait and psychiatric comorbidity of our patients. Previous studies have in fact found that these variables can influence VEP responses [26,27]”.

Comment 12.  The correlation analyses between VEP parameters and clinical variables should be presented more thoroughly, including correlation coefficients and p-values.

Response 12: We have now included all correlations in a new Table 3.

Comment 13.  While the study adds to the body of research on MOH, its findings regarding sensitization and habituation do not significantly differ from previous studies. The authors should emphasize more clearly how their findings advance existing knowledge, especially given the overlap with prior results​.

Response 13: Added as requested, as follows:

“Furthermore, as the learning phenomena of habituation and sensitisation are elementary forms of activity-dependent synaptic plasticity [25], we believe our results advance existing knowledge in that they show an imbalance of synaptic mechanisms of short-term potentiation and depression slightly in favour of the latter in subjects with MOH. From the analysis of our data, it is not possible to speculate whether this result reflects a primary dysfunction of cortical arousal mechanisms in these patients or is a secondary protective mechanism to counteract overstimulation of sensory systems by symptomatic drugs. Our research indicates that, the mechanisms of short-term plasticity triggered by the recurrent presentation of visual stimuli are partially impaired in MOH patients compared to pure CM patients and healthy volunteers. The subtle divergent plasticity mechanisms observed in the two patient groups may suggest that MOH and CM, although presenting a comparable phenotype, engage in distinct neurophysiological learning processes, potentially linked to divergent pathophysiological mechanisms underlying migraine chronification. Moreover, although we did not identify an association between the consumption of acute drugs and the electrophysiological parameters, persistent exposure to acute medication usage may lead to modifications in short-term plasticity mechanisms.”

Comment 14.  The study does not fully explore limitations such as the inability to prevent medication intake on the day of recordings for MOH patients. This could have influenced the VEP measurements and should be discussed more thoroughly.

Response 14: Added as study’s limitation as follows: “A further drawback of our investigation is the impossibility to restrict acute medication consumption on the day of recordings for MOH patients. This is due to the recognized compulsive behaviour and anticipatory anxiety exhibited by these patients [3,28].”.

Comment 15.  The discussion could benefit from a more in-depth exploration of potential mechanisms underlying any observed differences (or lack thereof) between CM and MOH patients.

Response 15: Added, also in response to the comments raised above.

Comment 16.  Conclusion: Please see my general recommendation: In order to put the gist of the manuscript in an appropriate manner, I would suggest that the review should be summarized in one single paragraph that should contain about 150-200 words, underlining that the authors represent the best experts in a broad and deep consideration of authors in this field. It would help because this subsequently supports the importance of their work to be able to clarify the theoretical indications and practical applications of its results. It is also important to mention aspects as regards potential lines of future research and others of a theoretical and methodological nature that it would be of prime importance to undertake in order for the significance of this line of study to be established in full.

Response 15: Following the Reviewer’s suggestion, now we revised to the conclusion as follows:

“This study revealed delayed visual information processing abnormalities verified by analysis of the P1-N2 component of VEPs only in MOH patients. Further investigations by independent research groups are necessary to ascertain whether this electrophysiological finding can serve as a biomarker for the disease, specifically in differentiating MOH from CM. The former will also assist in personalizing the available pharmacological and non-pharmacological interventions. In the MOH group of patients, no significant differences in amplitude and habituation of VEPs were found in subgroups of overusers and after abrupt acute medication discontinuation, compared to healthy subjects. Additional electrophysiological investigations are required to ascertain whether particular personality traits, the existence of systemic and central diseases in comorbidity, or unique genetic variants can variably influence sensory processing in CM and MOH patients.”

Comment 17.  The conclusion lacks a clear discussion of the clinical relevance of VEP findings, particularly in distinguishing CM from MOH for treatment purposes. Clarifying the practical implications of the results would enhance the paper’s impact.

Comment 18.  Finally, I believe that the clinical implications of the findings should be elaborated upon. How might these results impact understanding or management of CM and MOH?

Responses 17 & 18: Following the request of the Reviewer, now we added the following words to the conclusion: “Further investigations by independent research groups are necessary to ascertain whether this electrophysiological finding can serve as a biomarker for the disease, specifically in differentiating MOH from CM. The former will also assist in personalizing the available pharmacological and non-pharmacological interventions.”

Comment 19.  Please cite more references. An article, such as this, typically cites more than 60-70 references.

Response 19: I agree with you. The initial version of the article included more quotations, but they were self-quotations, and the Editorial Office asked me to remove them. Anyway, now with the addition of several further paragraphs, we have added more references to the list.

Reviewer 2 Report

Comments and Suggestions for Authors

Its great work, but there are some points need to explain:

1-    Title need to change to match with paper presentation (because you use MOH as a variable not CM

2-   In introduction the statement (Little is known about the 35 direct effect of acute medications at the cortical level.) without reference and also (Therefore, further research in a larger cohort of patients is needed to verify whether vis- 58 ual responses are different in patients with MOH and CM compared to healthy subjects.)

3-   19 were excluded because they did not fulfil the inclusion criteria (where are the inclusion criteria ????)

4-   In study participants, I think the range of age 18 to 65 years is too big

5-   Recording of Visual Evoked Potentials (VEP) need to be more concise

6-   Discussion should follow the usual presentation in clear & concise way

7-   Where the ethical commit & clinical trial registration number

8-   References need to be update   

Comments on the Quality of English Language

Need MINOR editing 

Author Response

Its great work, but there are some points need to explain:

Comment 1.    Title need to change to match with paper presentation (because you use MOH as a variable not CM

Response 1: We thank the Reviewer. Now, also in agreement with Reviewer 1 we have modified the manuscript’s title as follows: “Decoding Visual Responses: Insights into Chronic Migraine and Medication Overuse Headache with Electrophysiological Analysis”

Comment 2.   In introduction the statement (Little is known about the 35 direct effect of acute medications at the cortical level.) without reference and also (Therefore, further research in a larger cohort of patients is needed to verify whether vis- 58 ual responses are different in patients with MOH and CM compared to healthy subjects.)

Response 2: We added the same reference review of Ashina et al Lancet Neurol 2023 for both statements.

Comment 3.  19 were excluded because they did not fulfil the inclusion criteria (where are the inclusion criteria ????)

Response 3: In the following statement: “Participants were included if they were between 18 and 65 years of age and had at least a 1-year clinical history of migraine.” However, to seek for clarity we modified the pinpointed statement as follows: “were excluded because of the inclusion and exclusion criteria”.

Comment 4.   In study participants, I think the range of age 18 to 65 years is too big

Response 4: We thank the Reviewer for the question. We must admit that this is the standard range for all the observational studies in migraine.

Comment 5.   Recording of Visual Evoked Potentials (VEP) need to be more concise.

Response 5: We significantly shortened the description of VEP recording by removing 6 lines.

Comment 6.   Discussion should follow the usual presentation in clear & concise way

Response 6: The discussion was deeply revised also in agreement with the comments raised by Reviewer 1.

Comment 7.   Where the ethical commit & clinical trial registration number.

Response 7: We added Ethical committee number at the end of the manuscript (RIF.CE 4839-2019). This is not a clinical trial.

Comment 8.   References need to be update.

Response 8: Updated. Now the number of references increased from 17 to 28.

Reviewer 3 Report

Comments and Suggestions for Authors

- useful study from leading researchers

- sample size estimation, exclusion criteria represebtativeness (diabetes, hypertension are common in migraine)

- consider potential interactions in the analysis

Author Response

Comment 1. useful study from leading researchers

Response 1: Thank you for your positive comment. We appreciated.

Comment 2. sample size estimation, exclusion criteria represebtativeness (diabetes, hypertension are common in migraine)

Response 2: We have added the coexistence of various systemic diseases (such as diabetes, systemic hypertension, and rheumatologic disorders) as exclusion criteria. Actually, sample size was not based on formal statistical calculation. Now, we added this relevant information in the Statistical analysis section of the manuscript as follows:

“Sample size calculation was not based on formal statistics but on previous literature [16,17]”.

Comment 3. consider potential interactions in the analysis

Response 3: We thank the Reviewer for his/her comment. Now, also following the comments of Reviewer 1 and 2, we have added more words in the Results and more explanations in the Discussion/Conclusion of the manuscript. We think that all of them will improve the understanding of potentials interactions.

Round 2

Reviewer 1 Report

Comments and Suggestions for Authors

30 September 2024 

The 2nd review report on the manuscript, titled ‘Visual evoked potentials sensitization/habituation mechanisms in chronic migraine patients with or without medication over-use’ by Coppola G et al., submitted to Journal of Clinical Medicine

Manuscript ID: jcm-3227845

 To Authors, 

I am pleased to see that the authors have attempted to address the issues raised in the previous session. Nevertheless, the revisions remain partial. Prior to publication, I respectfully request that the authors consider my comments and revise the manuscript to meet the high standards of the journal. 

Please consider the following comments:

1.     Please show them in the manuscript: Please include ten keywords from Medical Subject Headings (MeSH) in the title and the first two sentences of the abstract.

2.     Please cite more references. An article, such as this, typically cites more than 60-70 references. 

Best regards,

Reviewer